*Corrigendum*

# Translational efficiency across healthy and tumor tissues is proliferation-related

Xavier Hernandez-Alias (iD), Hannah Benisty (iD), Martin H Schaefer (iD) & Luis Serrano (iD)

**Correction to:** *Mol Syst Biol.* (2020) 16: e9275. DOI 10.15252/msb.20199275 | Published online 9 March 2020

The authors noticed a mistake in the computation of wobble-base pairing rules of tRNAs with their respective codons, which affects some of the results. The mistake arises from a misinterpretation of the documentation of the tAI software by dos Reis *et al* (2003, 2004) (https://github.com/mariodosreis/tai), which led them to place STOP codons at the wrong position of the input list.

As a result of this mistake, the codon–anticodon pairing rules were often incorrectly established, and therefore, codon efficiency estimates were partly affected.

Although this error does not change the main conclusions, it is important that the corrected results are reported. The authors apologize for these errors and any confusion they may have caused.

In summary, the following items were affected by the error and are now corrected:

- Results, section "**tRNA repertoires determine tissue-specific translational efficiency**": Figure 3, Figure EV4, Table EV6, and Table EV7 have been updated. The originally published Table EV6 and Table EV7 are available with the Corrigendum. All comparisons remain significant, and therefore, the interpretation of the results is not affected.
- Results, section "**Aberrant translational efficiencies drive tumor progression**": Instead of ProCCA and GlyGGT reported initially, codons ArgAGA and ThrACT are the most significantly changing codons between healthy and tumor tissues. However, the main message of the section remains unaffected. The authors still detect proliferation-related functions being translationally affected in cancer. Figure 4, Table EV8, Table EV9, and Table EV10 have been updated. The originally published Table EV8, Table EV9 and Table EV10 are available with the Corrigendum. Moreover, the subsequent interpretations of these results are also corrected. Specifically:

  ○ Results, section "**Promoter methylation and gene copy number regulate the tRNA abundance**": Although the results of this section are not affected, the specific mention to ProCCA has lost its relevance because of changes in section "**Aberrant translational efficiencies drive tumor progression**". Updated Figures 5 and EV6 now show changes in ArgAGA instead.
  ○ Abstract, Synopsis, and Introduction: The specific mention of ProCCA became irrelevant. ProCCA is replaced by ArgAGA.
  ○ Discussion: ArgAGA is now discussed instead of ProCCA.

The specific corrections in the text are detailed below.

**Abstract**

**From:**

Furthermore, the aberrant translational efficiency of some codons in cancer, exemplified by ProCCA and GlyGGT, is associated with poor patient survival.

**To:**

Furthermore, the aberrant translational efficiency of some codons in cancer, exemplified by ArgAGA, is associated with poor patient survival.

**Synopsis, third bullet point**

**From:**

- Proliferation is the major determinant of translational efficiency differences among tissues, and the codon ProCCA appears particularly favored in cancer.

**To:**

- Proliferation is the major determinant of translational efficiency differences among tissues, and the codon ArgAGA appears particularly favored in cancer.

**Introduction**

**From:**

We discover multiple codons, including ProCCA and GlyGGT, whose translational efficiency is compromised and leads to poor prognosis in cancer.

**To:**

We discover multiple codons, including ArgAGA, whose translational efficiency is compromised and leads to poor prognosis in cancer.

**Results, section "tRNA repertoires determine tissue-specific translational efficiency"**

**From:**

When analyzing the tissue medians of SDA weights per each codon (SDAw), we observe that most codons are optimally balanced (SDAw = 1), while 12.4 and 23.6% of codons are favored (SDAw > 2) and disfavored (SDAw < 0.5), respectively.

**To:**

When analyzing the tissue medians of SDA weights per each codon (SDAw), we observe that most codons are optimally balanced (SDAw = 1), while 13.7 and 16.3% of codons are favored (SDAw > 2) and disfavored (SDAw < 0.5), respectively.

**From:**

Both the first and second components significantly correlate with the proliferation marker Ki67 (0.4 and 0.35; see Fig 3B). In agreement with the proliferation- and differentiation-related codons of

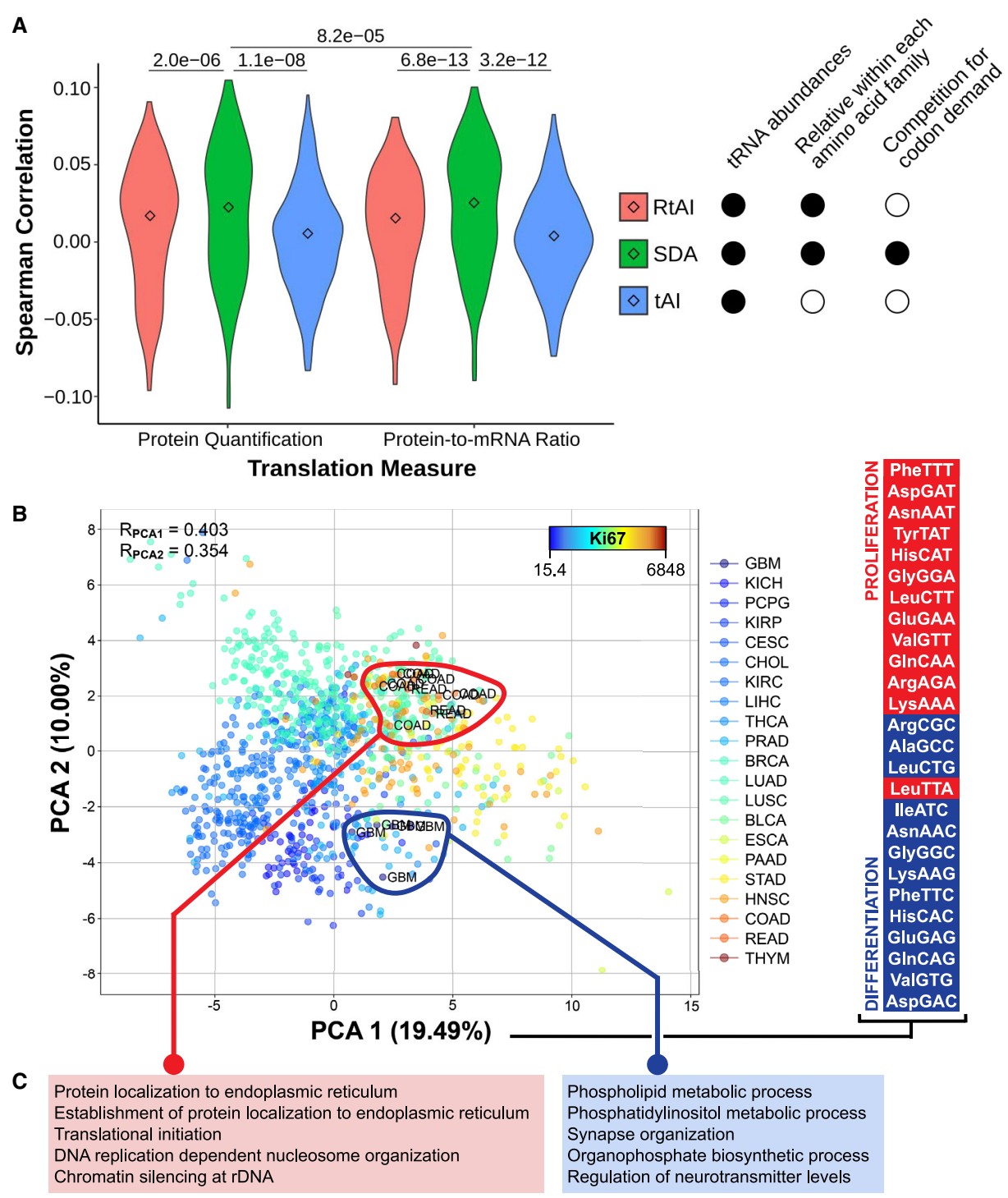

**Figure 3. Original.**

Gingold *et al* (2014), such proliferative pattern is similarly reproduced by the codons contributing to the first PCA component, which has the strongest association to proliferation (Fig 3B).

**To:**

Both the first and second components significantly correlate with the proliferation marker Ki67 (0.4 and -0.24; see Fig 3B). In agreement with the proliferation- and differentiation-related codons of

Gingold *et al* (2014), such proliferative pattern is similarly reproduced by the codons contributing to the first (Fig 3B) and second (Table EV6) PCA components.

**From:**

Consistent with our hypothesis, the results indicate that gut-optimized proteins are enriched in translation, DNA replication, and protein localization, whereas brain-optimized proteins are

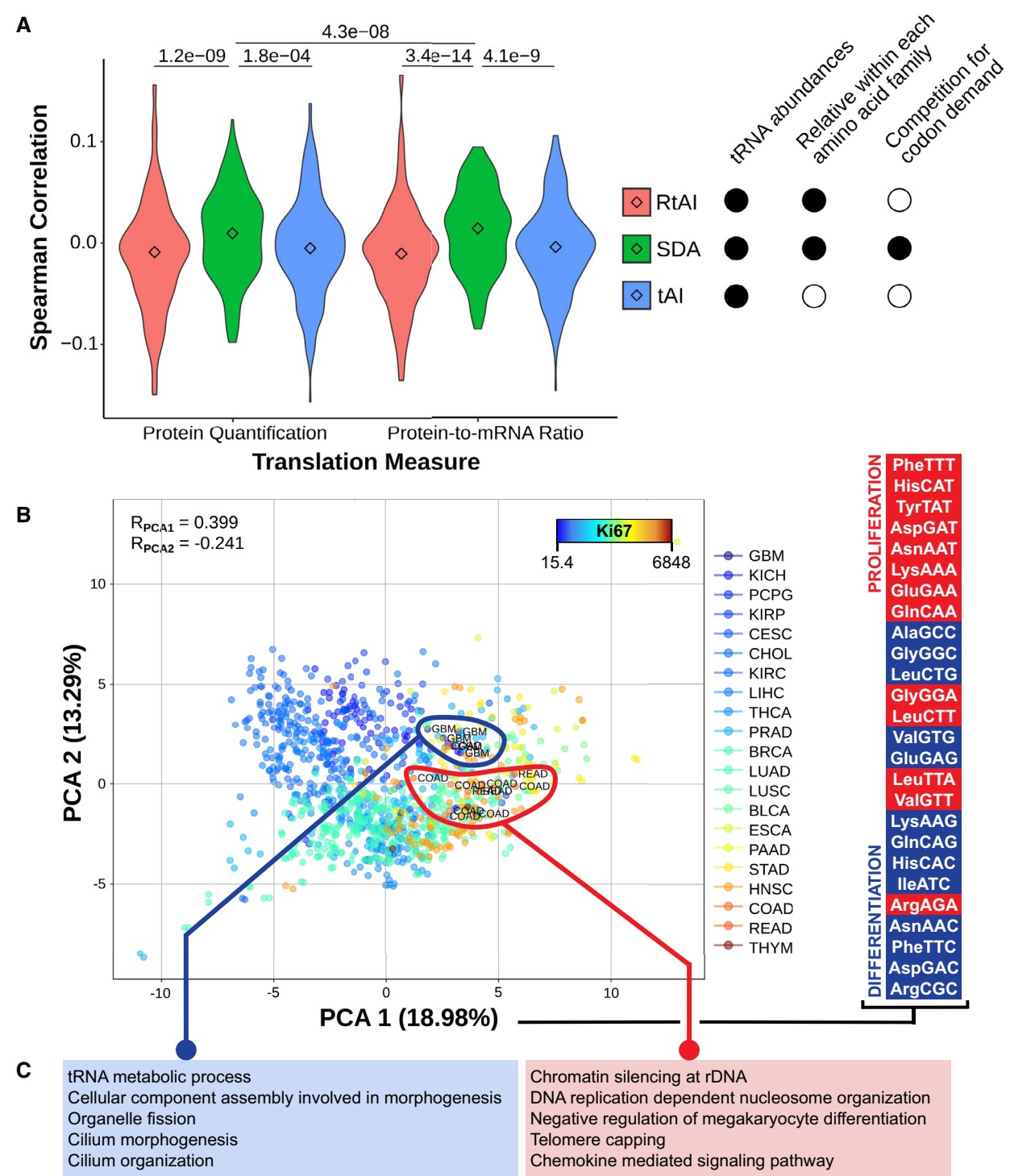

Figure 3. Corrected.

related to phospholipid production and neural function (Fig 3C, Table EV7).

**To:**

Consistent with our hypothesis, the results indicate that gut-optimized proteins are enriched in DNA replication, chromatin organization, and chemokine signaling, whereas brain-optimized proteins are related to tRNA metabolism and cilium morphogenesis (Fig 3C, Table EV7).

**Results, section "Aberrant translational efficiencies drive tumor progression"**

**From:**

Among the most consistent changes, the ProCCA codon is significantly more favored in tumors for 8 out of 10 cancer types, while the ProCCG is disfavored in 14 out of 16 cancers (Fig 4B). In the case of glycine, GlyGGT is better adapted in healthy

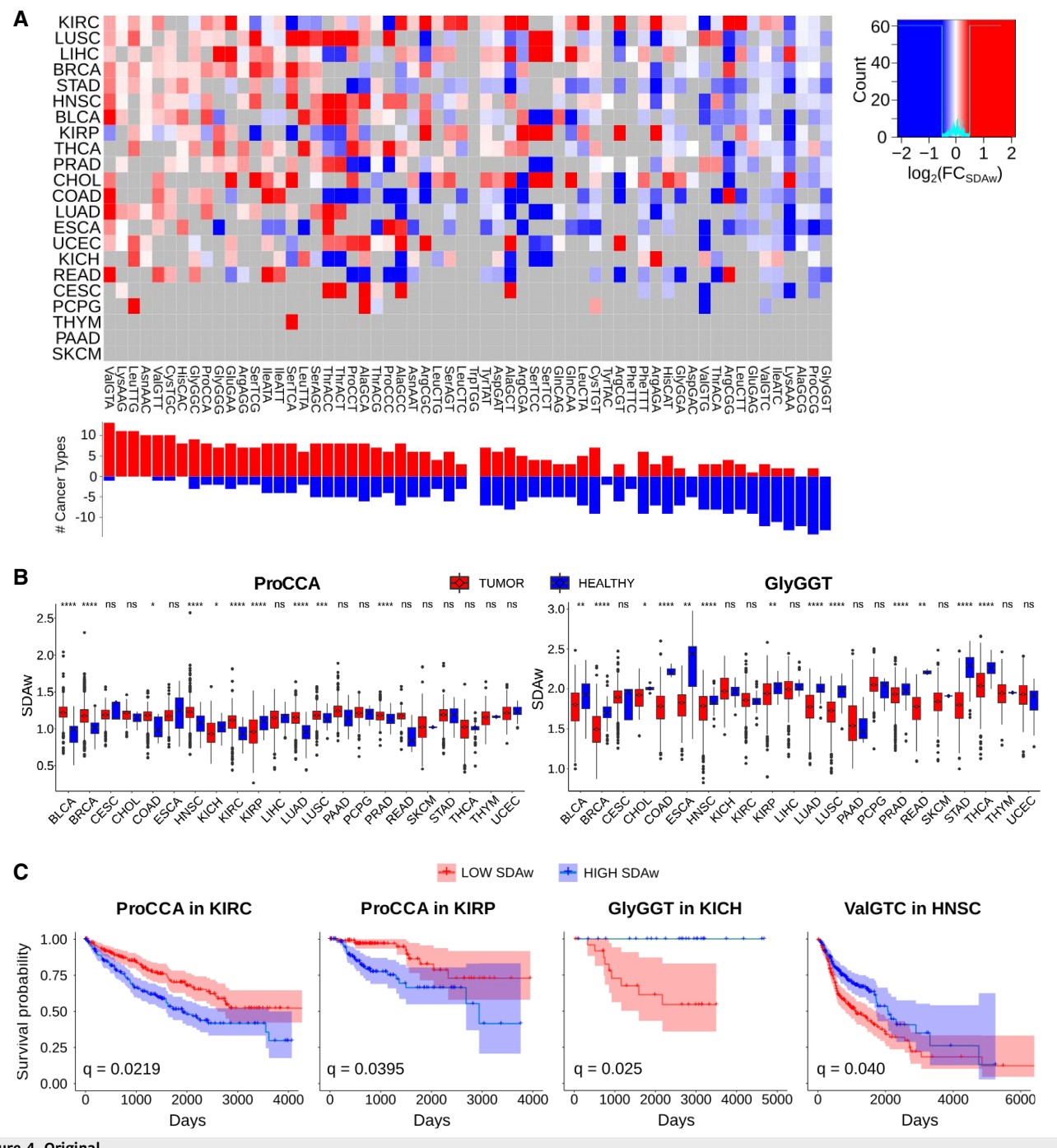

Figure 4. Original.

samples (13/13), whereas tumor mostly favors GlyGGC (9/12) and GlyGGG (7/9).

**To:**

Among the most consistent changes, the ArgAGA codon is significantly more favored in tumors for 15 out of 15 cancer types, while the ArgCGG is disfavored in 7 out of 11 cancers (Fig 4B). In the case of threonine, ThrACT and ThrACC are better adapted in healthy samples (13/14), whereas tumor mostly favors ThrACG (12/16).

**From:**

Among others, and consistent with the previous analysis, high supply-to-demand weights of ProCCA are associated with poor prognosis in kidney renal clear cell carcinoma and kidney renal papillary cell carcinoma. Proline limitation in clear cell renal cell carcinoma has been shown to compromise CCA-decoding tRNAPro aminoacylation, leading to reduced tumor growth (Loayza-Puch et al, 2016). In contrast, high SDAw of GlyGGT and ValGTC lead to longer survival in kidney chromophobe and head and neck squamous cell carcinoma, respectively.

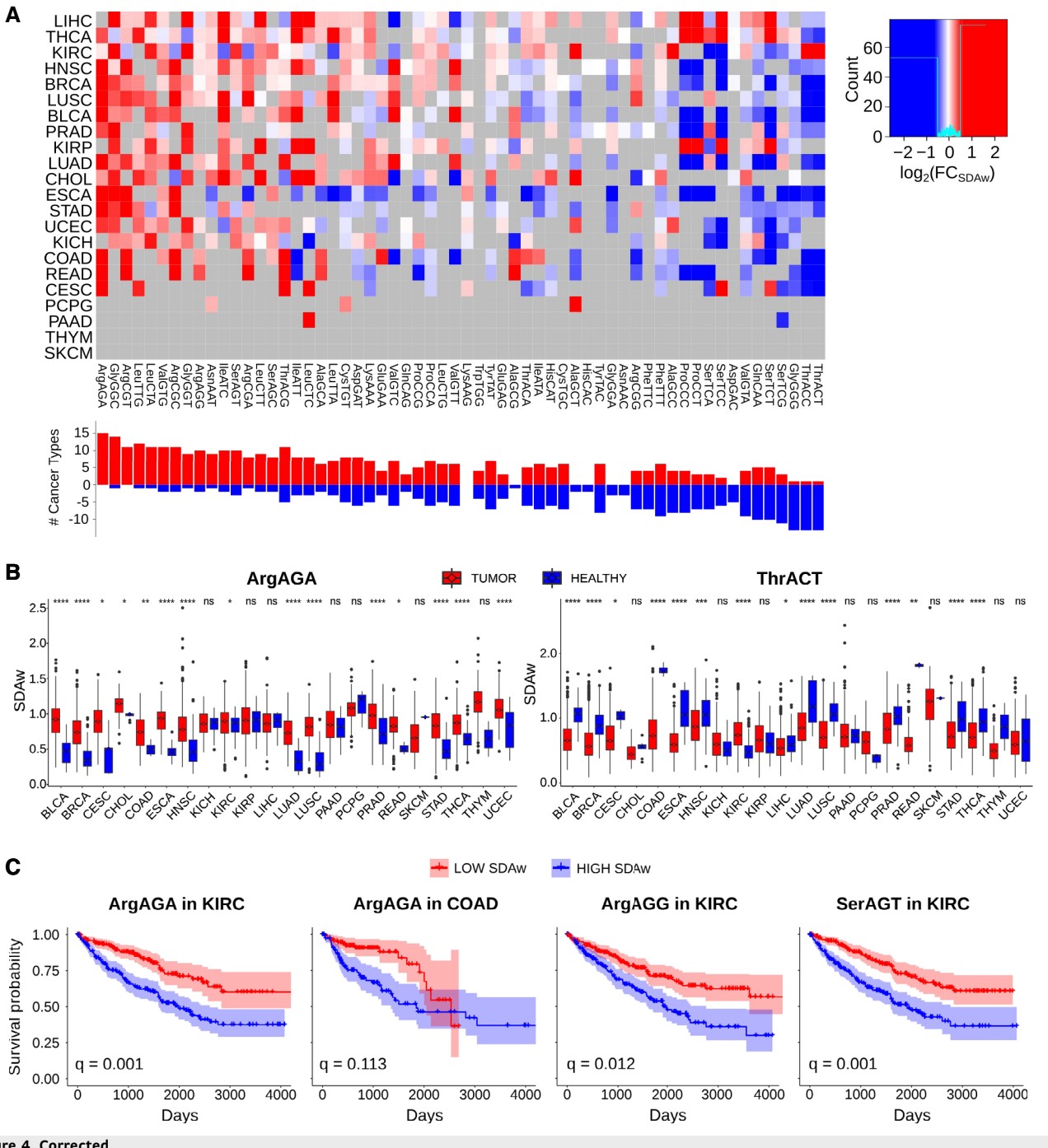

**Figure 4. Corrected.**

**To:**

Among others, and consistent with the previous analysis, high supply-to-demand weights of ArgAGA are associated with poor prognosis in kidney renal clear cell carcinoma and colon adenocarcinoma. Arginine limitation in the kidney cell line HEK293T has been shown to compromise tRNAArg aminoacylation, leading to codon pausing and reduced cell viability (Darnell *et al*, 2018). In addition, low SDAw of ArgAGG and SerAGT lead to longer survival in kidney renal clear cell carcinoma.

**From:**

To determine the impact of aberrant translational efficiencies in regulating an oncogenic translation program, we calculate the differential SDA for the whole genome based on the average SDAw of healthy and tumor samples in kidney renal clear cell carcinoma, since it is the cancer type with the most SDAw differences (Fig 4A). The GSEA of the resulting ΔSDA score indicates that cancer SDAw should favor the translation of proteins related to DNA replication and gene expression, whereas the healthy kidney samples favor

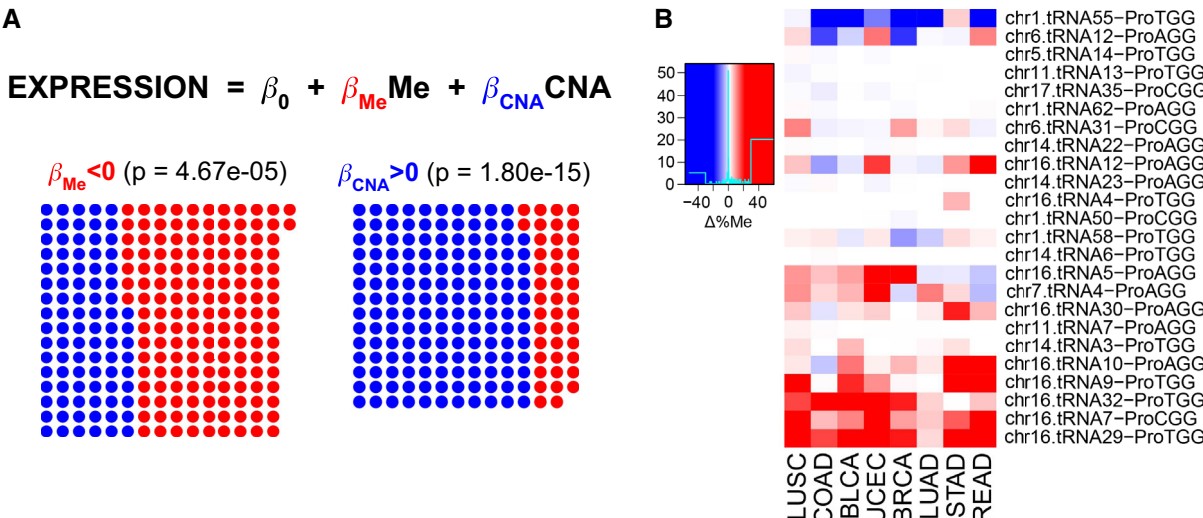

**Figure 5. Original.**

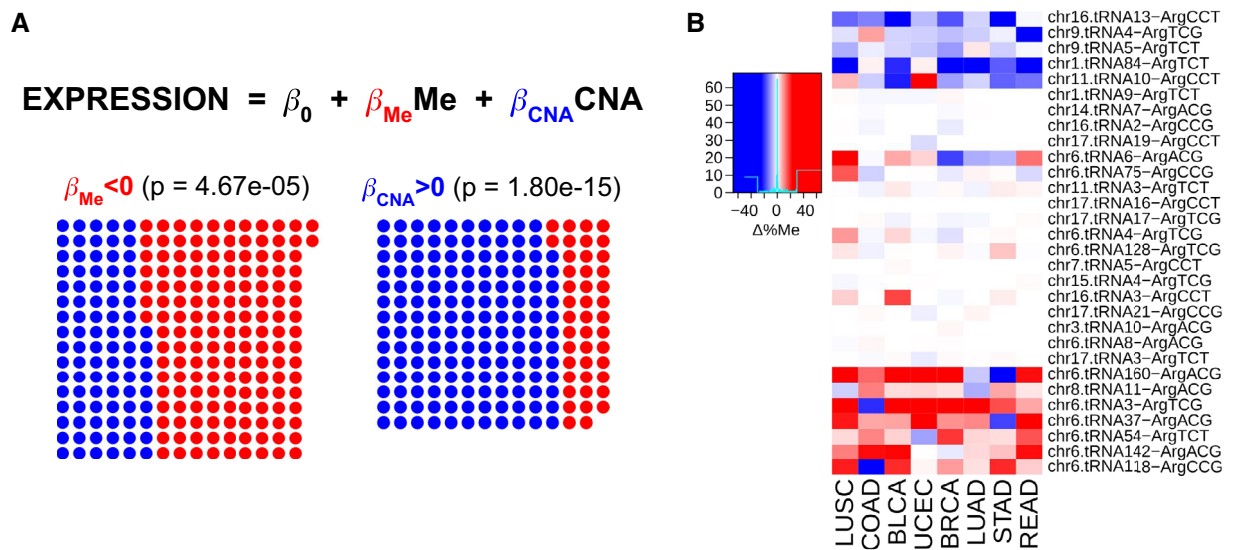

**Figure 5. Corrected.**

development and differentiation processes (Table EV9). As the SDAw of the ProCCA is specifically disturbed in cancer, we also interrogate how this codon is distributed along the genome. We therefore perform a GSEA on the relative codon usage of ProCCA, which shows that DNA replication and cell cycle functions lie among the most CCA-enriched genes, while morphogenesis and differentiation terms are CCA-depleted (Table EV10).

**To:**

To determine the impact of aberrant translational efficiencies in regulating an oncogenic translation program, we calculate the differential SDA for the whole genome based on the average SDAw of healthy and tumor samples in kidney renal clear cell carcinoma, since it is the cancer type with the most prognostic differences (Fig 4A). The GSEA of the resulting ΔSDA score indicates that cancer SDAw should favor the translation of proteins related to DNA replication and gene expression, whereas the healthy kidney

samples favor signals transduction and differentiation processes (Table EV9). As the SDAw of the ArgAGA is specifically disturbed in cancer, we also interrogate how this codon is distributed along the genome. We therefore perform a GSEA on the relative codon usage of ArgAGA, which shows that proliferation and immune activation functions lie among the most AGA-enriched genes, while development and differentiation terms are AGA-depleted (Table EV10).

**From:**

In particular, ProCCA appears as an interesting codon candidate in favoring tumor progression, which we had also detected in healthy tissues to be associated with proliferation (Fig 3B, Table EV6).

**To:**

In particular, ArgAGA appears as an interesting codon candidate in favoring tumor progression, which we had also detected in healthy tissues to be associated with proliferation (PCA2 in Table EV6).

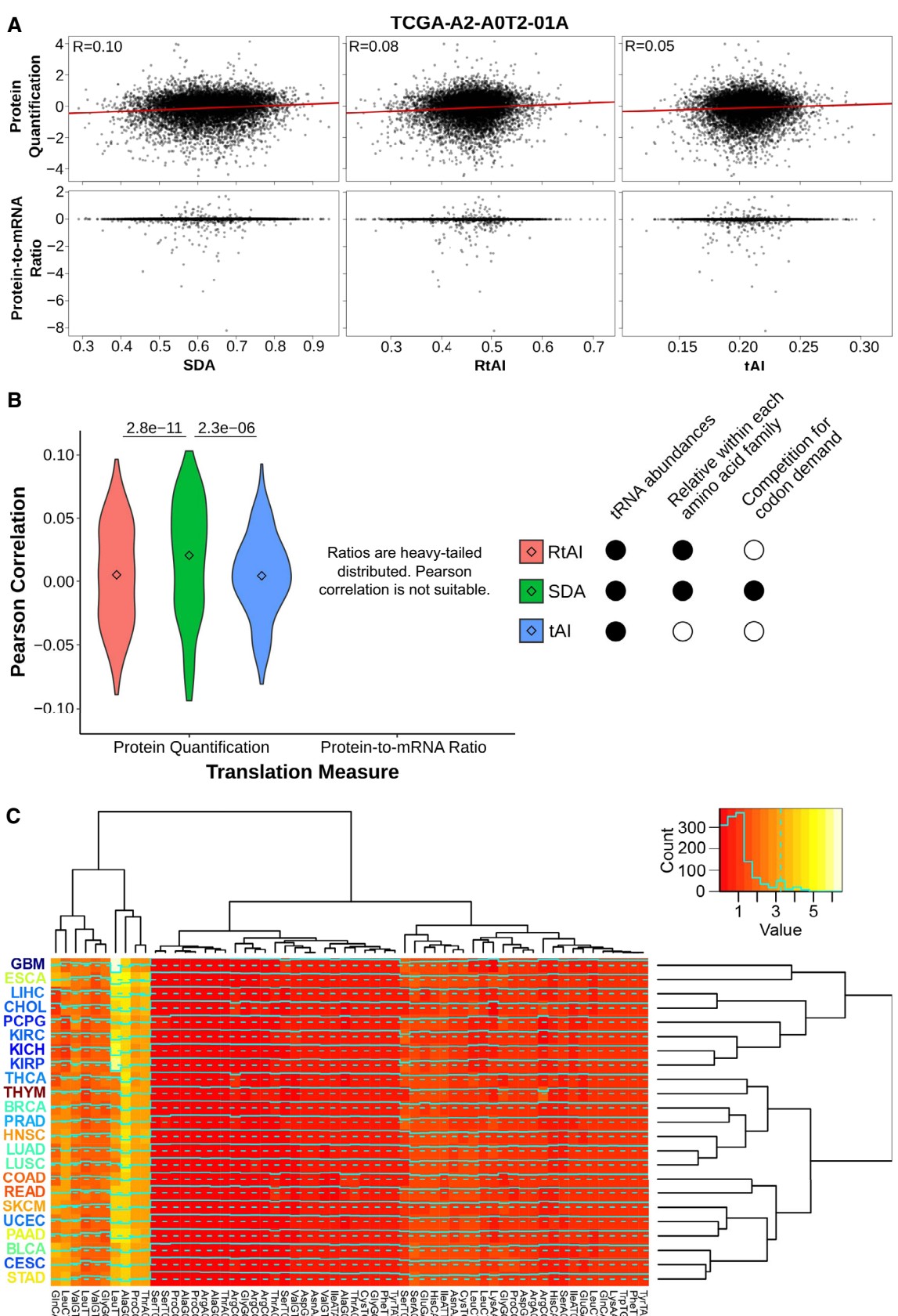

**Figure EV4. Original.**

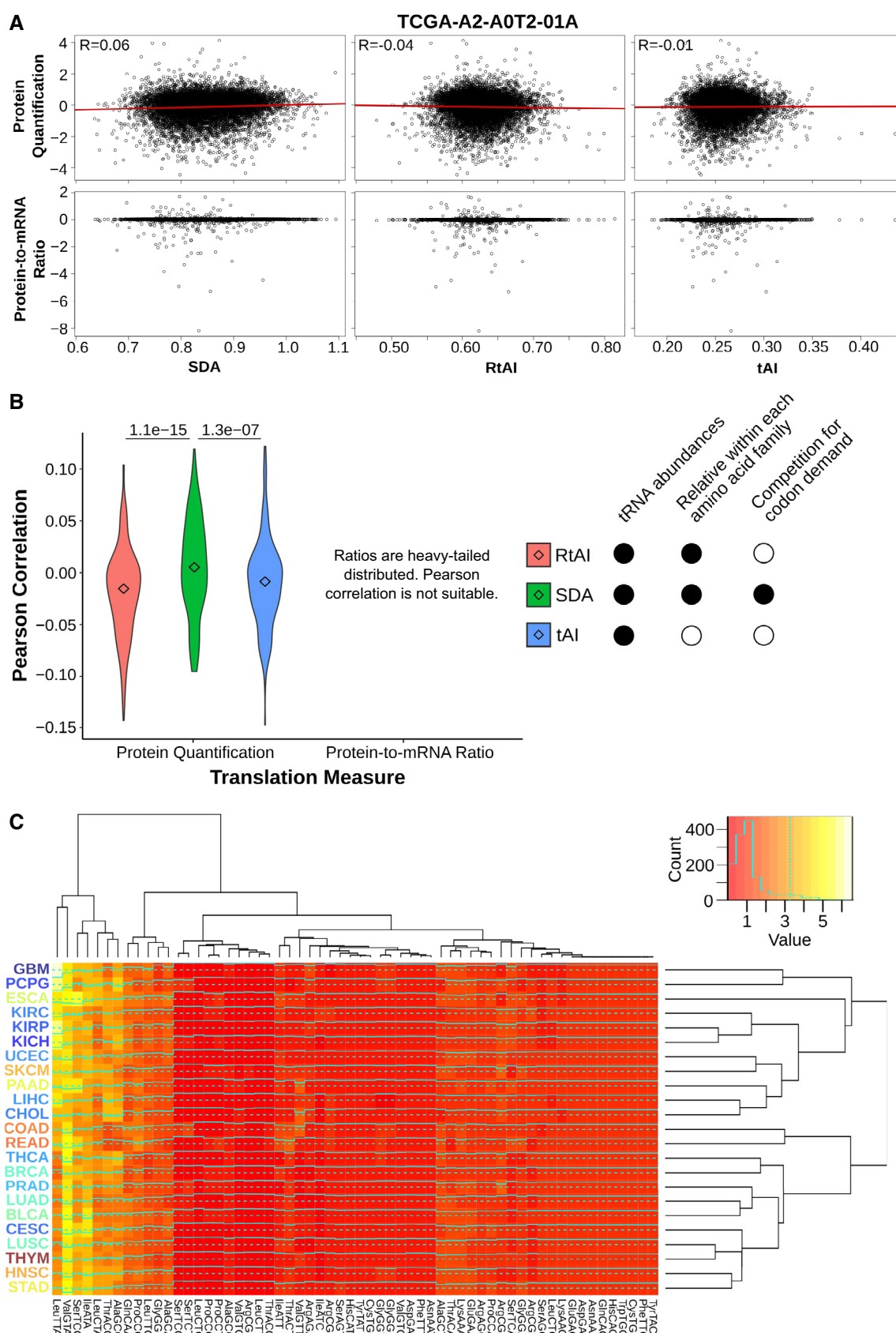

**Figure EV4. Corrected.**

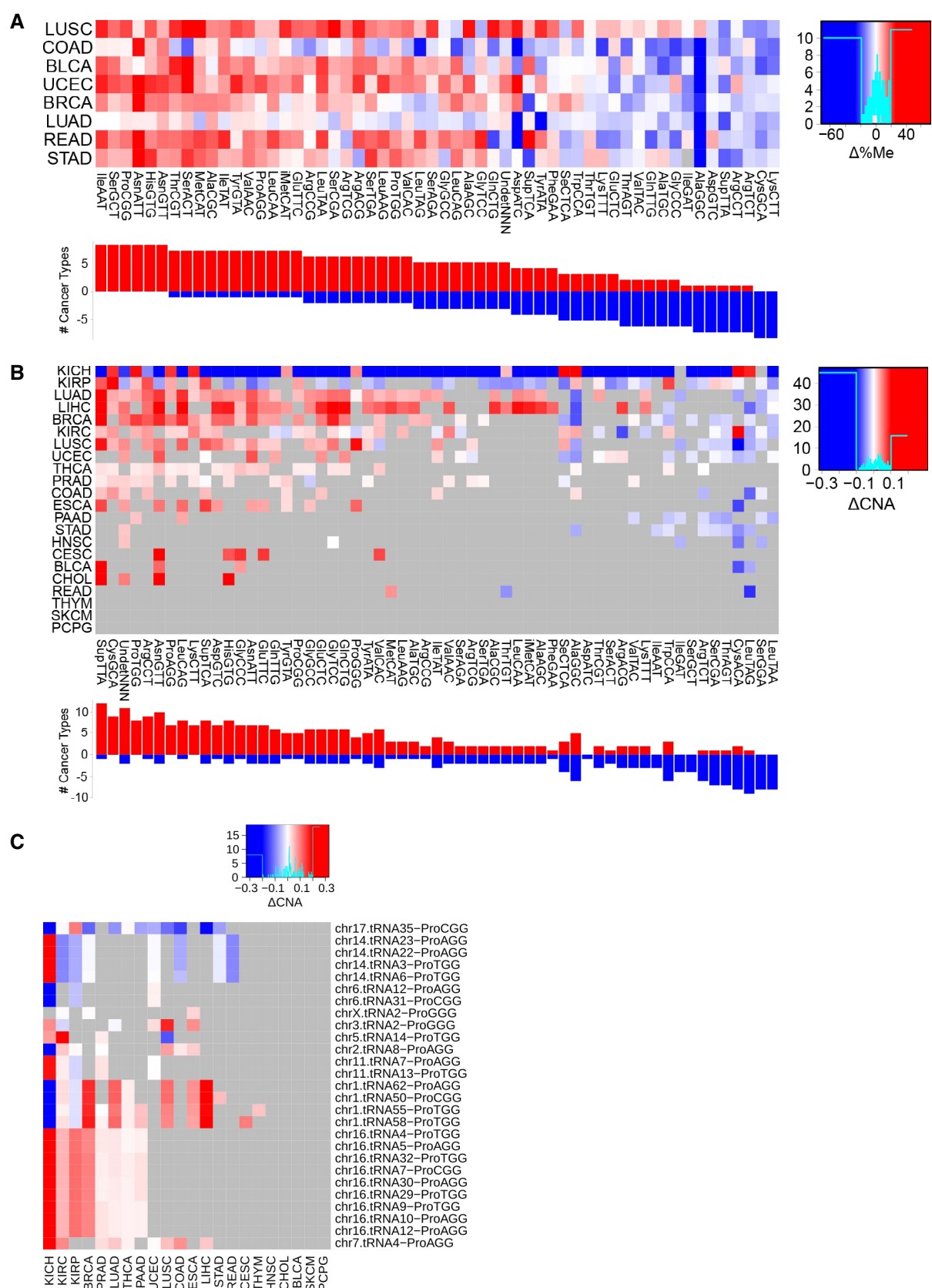

Figure EV6. Original.

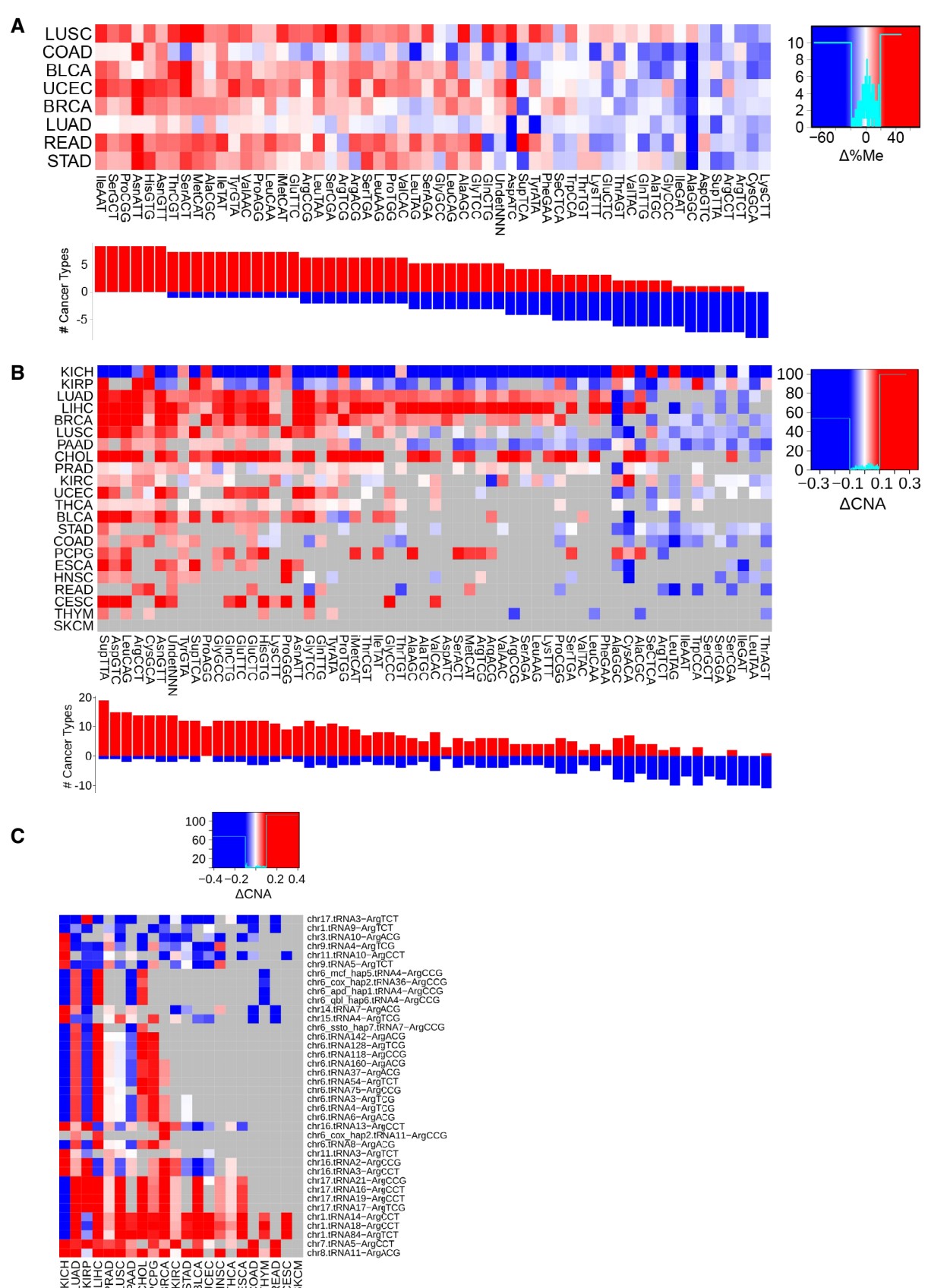

Figure EV6. Corrected.

**Results, section "Promoter methylation and gene copy number regulate the tRNA abundance"**

**From:**

Given the association of the codon ProCCA with cancer prognosis (Fig 4C), we explore the abundance pattern of tRNAPro in TCGA. While both tRNAProTGG and tRNAProAGG are able to decode ProCCA, the latter specifically appears overexpressed in 8 out of 9 cancer types (Fig EV5A), making it a candidate driver of the translational differences.

**To:**

Given the association of the codon ArgAGA with cancer prognosis (Fig 4C), we explore the abundance pattern of tRNAArg in TCGA. In agreement, the complementary tRNAArg$^{TCT}$ appears overexpressed in 13 out of 15 cancer types (Fig EV5A), making it a candidate driver of the translational differences.

**From:**

In total, tRNAPro$^{AGG}$ genes stand among the most duplicated and least methylated proline isoacceptors in cancer (Fig EV6A and B), in particular at the chr6.tRNA12 and chr16.tRNA12 genes (Fig 5B).

**To:**

In total, tRNAArg$^{TCT}$ genes stand among the least methylated arginine isoacceptors in cancer (Fig EV6A and B), in particular at the chr9.tRNA5 and chr1.tRNA84 genes (Fig 5B).

**Discussion**

**From:**

In particular, we detect the ProCCA codon to be significantly more favored in proliferative cells and leading to poor cancer prognosis in kidney carcinomas, specifically driven by an overexpression of tRNAPro$^{AGG}$ in cancer. Proline limitation in clear cell renal cell carcinoma has indeed been shown to mostly compromise tRNA-Pro$^{AGG}$ aminoacylation, leading to slower proline translation and reduced tumor growth (Loayza-Puch *et al*, 2016). Furthermore, in support of our approach for isoacceptor quantification and translational efficiency, similar studies of tRNA levels in TCGA have controversially claimed an opposite prognostic value for the ProCCA codon in clear renal cell carcinoma (Zhang *et al*, 2018, 2019).

**To:**

In particular, we detect the ArgAGA codon to be significantly more favored in proliferative cells and leading to poor cancer prognosis in kidney carcinoma, specifically driven by an overexpression of tRNAArg$^{TCT}$ in cancer. Arginine limitation in the kidney cell line HEK293T has indeed been shown to compromise tRNAArg aminoacylation, leading to arginine codon pausing and reduced cell viability (Darnell *et al*, 2018). Furthermore, in support of our approach for isoacceptor quantification and translational efficiency, similar studies of tRNA levels in TCGA have concordantly claimed a prognostic value for the ArgAGA codon in clear renal cell carcinoma (Zhang *et al*, 2018, 2019).

**From:**

Here, we specifically propose a role for DNA methylation in regulating the overexpression of tRNAPro$^{AGG}$ in cancer, although no direct causal link can yet be established.

**To:**

Here, we specifically propose a role for DNA methylation in regulating the overexpression of tRNAArg$^{TCT}$ in cancer, although no direct causal link can yet be established.

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

**Figure 3C legend**

**From:**

GSEA of the differential SDA between extreme tissues ($\Delta$SDA = SDA$_{Colorectal}$ - SDA$_{Brain}$), showing the top five GO terms with high (left) and low (right) SDA in colorectal versus glial tissues.

**To:**

GSEA of the differential SDA between extreme tissues ($\Delta$SDA = SDA$_{Colorectal}$ - SDA$_{Brain}$), showing five among the top ten GO terms with high (right) and low (left) SDA in colorectal versus glial tissues.

**Figure 4B legend**

**From:**

Boxplot of the SDAw of ProCAA and AlaGCG codons across TCGA cancer types.

**To:**

Boxplot of the SDAw of ArgAGA and ThrACT codons across TCGA cancer types.

**Figure 4C legend**

**From:**

Survival curves for the previous codons in KIRC, KIRP, and BLCA patients.

**To:**

Survival curves for the previous codons in KIRC and COAD patients.

**Figure 5B legend**

**From:**

Differential promoter methylation (bisulfite sequencing) between healthy and tumor samples of genes expressing proline tRNAs, as measured by $\Delta\%Me=(\%Me_{Tumor}- \%Me_{Healthy})$.

**To:**

Differential promoter methylation (bisulfite sequencing) between healthy and tumor samples of genes expressing arginine tRNAs, as measured by $\Delta\%Me=(\%Me_{Tumor}- \%Me_{Healthy})$.

**Figure EV6C legend**

**From:**

Differential gene copy number between healthy and tumor samples of genes expressing proline tRNAs, as measured by $\Delta$CNA.

**To:**

Differential gene copy number between healthy and tumor samples of genes expressing arginine tRNAs, as measured by $\Delta$CNA.

**Table EV10 description**

**From:**

Table EV10. GSEA RCU ProcCCA.

**To:**

Table EV10. GSEA RCU ArgAGA.

