## [Reviewer Comments PDF · Molecular Systems Biology]

Translational efficiency across healthy and tumor tissues is proliferation-related

Xavier Hernandez-Alias, Hannah Benisty, Martin H Schaefer, Luis Serrano

ORIGINAL PUBLICATION: Mol Syst Biol (2020) 16: e9275
DOI: 10.15252/msb.20199275

Correction timeline:

Received:	4 November 2020
Editorial communication:	18 November 2020
Editorial decision:	15 January 2021

Staff: Maria Polychronidou, Bernd Pulverer, Erica Wilfong Boxheimer

Report:

Editorial communication

18 November 2020

I am contacting you regarding the study by Luis Serrano and colleagues (MSB-19-9275 'Translational efficiency across healthy and tumor tissues is proliferation-related'), which you reviewed some time ago for Molecular Systems Biology. The authors have contacted us mentioning that after publication they detected a mistake in the code that affects some of the results. The edited article with their proposed corrections is attached below.

In brief, the authors state that the error is related to the computation of wobble-base pairing rules of tRNAs with their respective codons and arises in part from the poor documentation of the tAI software by dos Reis et al. (2003, 2004). They say that the software documentation on GitHub describes that, to compute the codon weights resulting from their interactions with tRNAs (get.ws function of the software), the abundances of all tRNAs need to be inputted in the specified order: "The tRNAs are ordered according to their anticodon complement, in the same order as in codonM's output as indicated above (but with added STOP codon complements)". However, it does not specify where exactly STOP codon complements need to be placed within the list of other codons. In the published version of the study, the authors had incorrectly added them at the end of the whole list, but they actually needed to be intercalated among the rest of codon complements. The correct order of codons (STOP codons in []) is as follows: TTT, TTC, TTA, TTG, TCT, TCC, TCA, TCG, TAT, TAC, [TAA], [TAG], TGT, TGC, [TGA], TGG, CTT, CTC, CTA, CTG, CCT, CCC, CCA, CCG, CAT, CAC, CAA, CAG, CGT, CGC, CGA, CGG, ATT, ATC, ATA, ATG, ACT, ACC, ACA, ACG, AAT, AAC, AAA, AAG, AGT, AGC, AGA, AGG, GTT, GTC, GTA, GTG, GCT, GCC, GCA, GCG, GAT, GAC, GAA, GAG, GGT, GGC, GGA, GGG.

As a result of the wrong intercalation of stop codons, the codon-anticodon pairing rules were often incorrectly established, and therefore the codon efficiency estimates are partly affected.

In sum this mistake affects the following:

- Section 3 of the Results. Figure 3, Figure EV4 and Table EV6-7 need to be updated. The authors state that nevertheless all comparisons remain significant.
- Section 4 of the Results. ProCCA and GlyGGT are NOT the most significantly changing codons between healthy and tumor tissues, but codons ArgAGA and ThrACT need to be considered instead.

The authors state that the main message of the section is not affected, since they continue detecting proliferation-related functions being translationally affected in cancer. This affects:
--- Figure 4, Table EV8-10 and their interpretations in Section 5 of the Results. Although the results of this section are not affected, the specific mention to ProCCA has lost the relevance because of changes in section 4. Figure 5 and Figure EV6 need to be updated to show changes of ArgAGA instead.
--- Abstract, Introduction and Discussion. Again, the specific mention of ProCCA has lost its relevance and needs to be replaced by ArgAGA.

The authors provide the amended manuscript, figures and tables and propose publishing a Corrigendum.

Since this error results in significant modifications and affects several conclusions, including that ArgAGA and NOT ProCCA shows altered translational efficiency in cancer, I would be grateful if you could take a look at the corrections that the authors propose and let us know whether you think that they seem acceptable or if in your opinion the main conclusions and findings are compromised.

Editorial decision

15 January 2021

Thank you again for alerting us to the errors in your recently published study and for providing a detailed explanation as well as all the corrected figures, datasets and text. I apologise for the delay in getting back to you, but we had to editorially discuss this at length, as it is a rather extensive correction. We also run it by reviewer #3, who confirmed that the error indeed does not change the main conclusions of the study and agreed that it is important to publish a Corrigendum to address the impact of the error. Taken together, we can now proceed with publishing the Corrigendum.

I have prepared the text for the Corrigendum (see attached file). Could you please confirm that all information in this text is correct so that we can forward it to production?

After transfer to production, our production team at Wiley will send you proofs within 2-3 weeks.

REFEREE REPORTS

Reviewer #3:

The manuscript in question uses small RNA sequencing data to quantify tRNA abundance and links changes in tRNA abundance with codon composition bias among target genes. The corrigendum addresses a technical error in determining the tRNA dependencies of different genes. This is an important point to correct, and in the new analysis, different tRNAs are seen to distinguish proliferative cancer cells from healthy controls. Broadly, corroborating evidence is provided to support the importance of the corrected tRNAs in cancer, although similar evidence existed to corroborate the tRNAs identified in the previous submission. The corrigendum does not alter the key conceptual conclusions of the manuscript, however, and addresses the specific impact of the technical error.